# Treatment with a GSK-3β/HDAC Dual Inhibitor Restores Neuronal Survival and Maturation in an In Vitro and In Vivo Model of *CDKL5* Deficiency Disorder

**DOI:** 10.3390/ijms22115950

**Published:** 2021-05-31

**Authors:** Manuela Loi, Laura Gennaccaro, Claudia Fuchs, Stefania Trazzi, Giorgio Medici, Giuseppe Galvani, Nicola Mottolese, Marianna Tassinari, Roberto Rimondini Giorgini, Andrea Milelli, Elisabetta Ciani

**Affiliations:** 1Department of Biomedical and Neuromotor Sciences, University of Bologna, Piazza di Porta San Donato 2, 40126 Bologna, Italy; manuela.loi3@unibo.it (M.L.); laura.gennaccaro@gmail.com (L.G.); claudiafuchs83@hotmail.com (C.F.); stefania.trazzi3@unibo.it (S.T.); giorgio.medici2@unibo.it (G.M.); giuseppe.galvani2@unibo.it (G.G.); nicola.mottolese@studio.unibo.it (N.M.); marianna.tassinari5@unibo.it (M.T.); 2Department of Medical and Clinical Sciences, University of Bologna, 40126 Bologna, Italy; roberto.rimondini@unibo.it; 3Department for Life Quality Studies, University of Bologna, 47921 Rimini, Italy; andrea.milelli3@unibo.it

**Keywords:** *CDKL5* deficiency disorder, GSK-3β, HDAC6, dual inhibitor, neuronal survival, hippocampal defects, synapse development

## Abstract

Mutations in the X-linked cyclin-dependent kinase-like 5 (*CDKL5*) gene cause a rare neurodevelopmental disorder characterized by early-onset seizures and severe cognitive, motor, and visual impairments. To date there are no therapies for *CDKL5* deficiency disorder (CDD). In view of the severity of the neurological phenotype of CDD patients it is widely assumed that *CDKL5* may influence the activity of a variety of cellular pathways, suggesting that an approach aimed at targeting multiple cellular pathways simultaneously might be more effective for CDD. Previous findings showed that a single-target therapy aimed at normalizing impaired GSK-3β or histone deacetylase (HDAC) activity improved neurodevelopmental and cognitive alterations in a mouse model of CDD. Here we tested the ability of a first-in-class GSK-3β/HDAC dual inhibitor, Compound **11** (**C11**), to rescue CDD-related phenotypes. We found that **C11**, through inhibition of GSK-3β and HDAC6 activity, not only restored maturation, but also significantly improved survival of both human *CDKL5*-deficient cells and hippocampal neurons from *Cdkl5* KO mice. Importantly, in vivo treatment with **C11** restored synapse development, neuronal survival, and microglia over-activation, and improved motor and cognitive abilities of *Cdkl5* KO mice, suggesting that dual GSK-3β/HDAC6 inhibitor therapy may have a wider therapeutic benefit in CDD patients.

## 1. Introduction

*CDKL5* deficiency disorder (CDD) is a complex and severe neurodevelopmental disorder caused by mutations of the *CDKL5* gene [1], for which a cure is not available. Patients with CDD are characterized by early-onset seizures and severe cognitive, motor, visual, and autonomic disturbances [1,2,3,4]. Incidence varies from 1:40,000 to 1:60,000 [5], and due to the fact that *CDKL5* is located on the X chromosome, the prevalence of CDD among women is four times higher than in men. Genetic mutations of the *CDKL5* gene cause absence of a functional CDKL5 protein, a serine/threonine kinase that is highly expressed in the brain and, in particular, in neurons [6,7].

In vitro and in vivo models of CDD have helped to provide important insights into the mechanism of CDKL5 functions in neuronal development. CDKL5 has been found to regulate neuronal migration, axon outgrowth, dendritic morphogenesis, and synapse development in cultured rodent neurons as a model system [8,9,10,11,12]. Similarly, *Cdkl5* deficiency in mice impairs spine maturation and dendritic arborization of hippocampal and cortical neurons [13,14,15,16,17,18], indicating that CDKL5 plays a role in dendritic morphogenesis and synapse development. CDKL5 has been shown to regulate cell survival [19,20,21]. *CDKL5* deletion in human neuroblastoma cells induces an increase in cell death and in DNA damage-associated biomarkers [19]. During brain aging, loss of *Cdkl5* was shown to decrease neuronal survival in different brain regions such as the hippocampus, cortex, and basal ganglia in *Cdkl5* knockout (KO) mice, paralleled by increased neuronal senescence [22].

In view of the variety of cellular processes regulated by protein kinases and of the severity of the neurological phenotype of CDD patients, it is widely assumed that CDKL5 may have a very complex role in neurons, influencing the activity of a variety of intracellular pathways. Indeed, through a kinome profiling study, Wang and colleagues demonstrated that several signaling transduction pathways involved in neuronal and synaptic plasticity are disrupted in the forebrain of *Cdkl5* KO mice [23]. Among the signaling pathways whose function is altered in the absence of *Cdkl5*, the AKT-GSK-3β signaling pathway is particularly interesting due to its pivotal role in brain development and function [24,25]. Increased GSK-3β activity plays a role in several neurodevelopmental alterations that characterize *Cdkl5*-deficient brains. Importantly, pharmacological inhibition of GSK-3β activity rescues dendritic morphogenesis and synapse development in the *Cdkl5* KO mouse [26,27]. Similar benefits to defective dendritic spine number and dynamics were obtained with the AKT activator IGF-1 in *Cdkl5* KO mice [17]. However, inhibition of GSK-3β activity has been shown to have positive effects in juvenile but not in adult *Cdkl5* KO mice [26], suggesting that pharmacological interventions aimed at normalizing only impaired GSK-3β activity might not be sufficient to restore the defects of a complex disease such as CDD. Interestingly, epigenetic modulators, including histone deacetylase (HDAC) inhibitors, have recently been shown to improve neurodevelopmental and cognitive alterations in *Cdkl5* KO mice [11], suggesting that imbalanced protein acetylation might represent a molecular mechanism that underlies Cdkl5 function.

Combinatorial therapies have recently become one of the most successful drug development strategies for complex diseases. For instance, it was reported that combined inhibition of GSK-3β and HDACs induces synergistic effects compared to the single target drug, with a potential improved therapeutic efficacy [28]. Due to the complexity of CDD, it is worth hypothesizing that the combined inhibition of GSK-3β and HDACs by a multi-target drug might be more efficient then a single-target therapy. Here, we examined the effect of treatment with a recently synthetized first-in-class GSK-3β/HDAC dual inhibitor, Compound **11** (**C11**) [29], to rescue CDD-related phenotypes in in vitro and in vivo models of CDD.

## 2. Results

### 2.1. Treatment with ***C11*** Restores GSK-3β and HDAC6 Activity in a Human Cellular Model of CDKL5 Deficiency

Compound **11** (**C11**) was selected for its high dual activity against GSK-3β and histone deacetylase 6 (HDAC6) [29], which offer potential nontoxic therapeutic targets for the amelioration of CNS development [30,31]. In order to confirm the dual inhibitory activity of **C11** on GSK-3β and HDAC6, we treated a recently generated human neuronal cell model of *CDKL5* deficiency, the *CDKL5* knockout (KO) SH-SY5Y neuroblastoma cell line (SH-*CDKL5*-KO; [19]), with **C11** (1 and 10 μM), and, as a control for selective GSK-3β inhibition, with NP-12 (Tideglusib; 1 μM).

GSK-3β is a constitutively active serine/threonine kinase that is predominantly modulated by inhibitory serine-9 (Ser9) phosphorylation [32]. Consistently with previous evidence [19], we found that phosphorylation of GSK-3β at Ser-9 was reduced in SH-*CDKL5*-KO compared with parental SH-SY5Y cells (Figure 1A,B). SH-*CDKL5*-KO cells treated with **C11** underwent an increase in Ser-9-phosphorylated GSK-3β levels that, at a dose of 10 μM, became similar to those of parental SH-SY5Y cells (Figure 1A,B). Similarly, NP-12-treated SH-*CDKL5*-KO cells showed an increase in GSK-3β phosphorylation (Figure 1A,B). As a molecular consequence of GSK-3β inhibition in SH-*CDKL5*-KO cells, we evaluated the levels of the downstream GSK-3β target, CRMP2 (collapsing response mediator protein 2), the phosphorylation of which is regulated by GSK-3β [33]. In line with an increased activity of GSK-3β (Figure 1A,B), we found higher phosphorylation levels of CRMP2 (Figure 1C,D) in SH-*CDKL5*-KO cells. Treatment with **C11** recovered CRMP2 phosphorylation in SH-*CDKL5*-KO cells (Figure 1C,D). As expected, we found a complete recovery of CRMP2 phosphorylation in NP12-treated SH-*CDKL5*-KO cells (Figure 1C,D).

Since tubulin is a substrate of HDAC6 [34], to confirm the inhibitory activity of **C11** on HDAC6, we investigated acetylated tubulin levels in untreated and treated SH-*CDKL5*-KO cells. Interestingly, we found that SH-*CDKL5*-KO cells showed decreased levels of acetylated tubulin compared with parental SH-SY5Y cells (Figure 1E,F), that were not related to increased HDAC6 levels (Appendix A). Treatment with **C11** recovered acetylated tubulin levels in SH-*CDKL5*-KO cells (Figure 1E,F). Importantly, the effects on tubulin were not observed when SH-*CDKL5*-KO cells were treated with NP12 (Figure 1E,F).

As previously demonstrated [29], the weak in vitro inhibitory activity of **C11** on other HDACs such as HDAC1 was confirmed by the lack of an effect of **C11** treatment on H3 acetylation levels in SH-*CDKL5*-KO cells (Appendix A).

### 2.2. Treatment with ***C11*** Restores Neuronal Maturation and Survival of a Human Cellular Model of CDKL5 Deficiency

When SH-SY5Y cells were treated with retinoic acid (RA) for 5 days, a considerable proportion of cells differentiated to a more neuronal phenotype by extending neuritic processes [19]. As previously reported [19], RA-treated SH-*CDKL5*-KO clones showed reduced neurite outgrowth in comparison with parental cells (Figure 2). Since **C11** and NP12 inhibit GSK-3β at different concentration ranges (**C11**: IC_50_ = 2.7 µM [29]; NP12: IC_50_ = 60 nM), to compare drug efficacy we used **C11** at a concentration that was 10 times higher (10 µM) than that of NP12 (1 µM). We found that treatment with **C11** restored neuritic length in SH-*CDKL5*-KO clones (Figure 2A,D). After treatment with NP-12, neuritic length in SH-*CDKL5*-KO cells increased to levels that were even higher than those of parental cells (Figure 2A,D), confirming the positive effect of GSK-3β inhibition in neuronal maturation [35].

To investigate cell proliferation and viability in SH-*CDKL5*-KO clones, we evaluated the percentage of mitotic and pyknotic nuclei visualized with Hoechst staining. As previously reported [19], SH-*CDKL5*-KO showed a reduced number of mitotic cells (Figure 2B,D) and an increase in the fraction of pyknotic nuclei (Figure 2C,D) compared to parental cells. Interestingly, we found that treatment with **C11** restored cell proliferation (Figure 2B,D) and strongly improved survival (Figure 2C,D) in SH-*CDKL5*-KO cells, while treatment with NP-12 had very limited or no effect (Figure 2B,D).

In parental SH-SY5Y cells, treatment with **C11** had no effect on neuronal maturation and survival (Appendix A).

Increased stress-induced cell death and DNA damage characterizes *CDKL5*-null cells [19]. To test whether **C11** protects SH-*CDKL5*-KO cells from stress, we examined cell viability and induction of DNA damage in SH-*CDKL5*-KO cells after oxidative (H_2_O_2_) stress. SH-*CDKL5*-KO cells were more sensitive to treatment with H_2_O_2_ than were parental SH-SY5Y cells, showing an increased number of pyknotic nuclei and γH2AX levels (Figure 2E,F). Treatment with **C11** drastically improved survival and DNA damage induction in SH-*CDKL5*-KO cells (Figure 2E,F). Again, treatment with NP-12 had no effect (Figure 2E,F). In agreement with the increased apoptotic cell death after oxidative stress, SH-*CDKL5*-KO cells showed decreased phosphorylation of AKT (Figure 2G) [36]. Restored levels of AKT phosphorylation in **C11**-treated, but not NP12-treated, SH-*CDKL5*-KO cells (Figure 2G), confirmed the pro-survival effect of **C11**.

### 2.3. Treatment with ***C11*** Restores Neuronal Maturation and Survival of Hippocampal Neurons from Cdkl5 KO Mice

As previously reported [11,12], hippocampal neurons generated from *Cdkl5* −/Y mice had a reduced neuritic length compared to control (+/Y) neurons (Figure 3), as well as a reduced number of dendritic PSD-95 (postsynaptic protein 95) immunoreactive puncta (Figure 3B,C). Treatment with **C11** or NP12 fully restored the reduced dendritic outgrowth of hippocampal neurons from *Cdkl5* −/Y mice (Figure 3A,C), suggesting that GSK-3β signaling plays a major role in hippocampal neuron maturation. In contrast, the number of PSD-95 immunoreactive puncta were not recovered by treatment with NP12 (Figure 3B,C) and only partially recovered by treatment with **C11** (Figure 3B,C).

It has been shown that differentiating hippocampal neurons generated from *Cdkl5* −/Y mice have a reduced survival rate [20], which is evident in the reduced number of *Cdkl5* −/Y neurons present in culture after 10 days of differentiation, in comparison with control (+/Y) cultures (Figure 3D). Treatment with **C11** restored neuronal survival in hippocampal cultures from *Cdkl5* −/Y (Figure 3D). Similarly to SH-*CDKL5*-KO cells, treatment with NP-12 had no effect on *Cdkl5* −/Y hippocampal neuron survival (Figure 3D).

In control (+/Y), neuron treatment with **C11** had no effect on neuronal maturation and survival (Appendix A).

### 2.4. Effect of Treatment with ***C11*** on Behaviors of Cdkl5 KO Mice

Based on the results obtained in vitro we sought to investigate whether dual GSK-3β and HDAC6 inhibition improves the neurodevelopmental alterations that characterize the *Cdkl5* −/Y brain. Since to date there are no studies on the action of **C11** in vivo, a pilot study was performed to examine the effect of an acute treatment with increasing **C11** doses (10, 50, and 100 mg/kg) on GSK-3β inhibition. We found a dose response increase of phosphorylated GSK-3β levels in both the hippocampus and cortex of *Cdkl5* −/Y mice (Figure 4A,B).

Based on these results, we treated 1-month-old *Cdkl5* −/Y mice (P30) daily for 15 days with **C11** (50 mg/kg), the lower dose that was shown to improve or even restore phosphorylated GSK-3β (P-GSK-3β) to control levels (Figure 4). Animals were behaviorally tested in the time window shown in Figure 5A and then sacrificed at P60. The effects of treatment on the neuroanatomy of the hippocampal region were examined in mice at P60. We found that treatment had no adverse effect on body weight, indicating that **C11** does not impair animals’ well-being (Figure 5B).

Common features of *CDKL5* deficiency disorder include decreased mobility, such as muscular rigidity, and repetitive hand movements (stereotypies), such as clasping and hand-sucking [37].

Catalepsy bar tests are widely used to measure the failure, resulting from muscular rigidity or akinesia, to correct an imposed posture [38]. Using this test we found that *Cdkl5* −/Y mice spent significantly more time hanging onto the bar than did control (+/Y) mice (Figure 5C). Treatment with **C11** restored this motor behavior in *Cdkl5* −/Y mice (Figure 5C), suggesting that treatment improves muscular rigidity due to loss of *C**dkl**5.*

Similarly to the human condition, *Cdkl5* KO mice exhibited a hind-limb clasping behavior [13]. In order to examine the effect of treatment with **C11** on motor stereotypies, we evaluated the hind-limb clasping time during the two minutes’ testing period (Figure 5D). While control (+/Y) mice exhibited very little hind-limb clasping, *C**dkl**5* −/Y mice spent more time in the clasping position (Figure 5D). Importantly, *C**dkl**5* −/Y mice treated with **C11** showed a significant decrease in clasping (Figure 5D), indicating a treatment-induced improvement of motor stereotypies.

Animals that had been treated with **C11** were subjected to the Morris Water Maze (MWM) test in order to explore hippocampus-dependent spatial memory. As previously reported [26], we found that, while control (+/Y) mice learned to find the platform by the second day, *Cdkl5* −/Y mice showed a clear learning deficit over the 2–4 days of the trial (Figure 5E). Treatment with **C11** improved the learning ability of *C**dkl**5* −/Y mice, which became significantly different from untreated *C**dkl**5* −/Y mice on the second and fourth day of testing (Figure 5E). In the probe test, as previously reported [26], *C**dkl**5* −/Y mice showed an increased latency to enter the former platform zone (Figure 5F). Treatment with **C11** in *C**dkl**5* −/Y mice caused a notable reduction in their latency to enter the former platform zone (Figure 5F), indicating an improvement in spatial memory. Performance in the MWM is influenced by motor functions that can be assessed by analyzing swimming patterns such as swim speed. No difference was observed between untreated and treated *C**dkl**5* +/Y and *C**dkl**5* −/Y mice as far as swim velocity was concerned (Figure 5G), indicating that the deficit of *C**dkl**5* −/Y mice in the hidden platform test was not caused by abnormalities in swimming abilities.

### 2.5. Effect of Treatment with ***C11*** on Synapse Development in Cdkl5 KO Mice

Alteration in synaptogenesis is now considered the main structural brain defect that underlies the behavioral abnormalities in the *C**dkl**5* KO mouse [11,17,27,39].

Given the positive effects of treatment with **C11** on behavior in *Cdkl5* −/Y mice, we investigated the effects of **C11** on spine maturation/connectivity. As previously reported [26], spine density of Golgi-stained CA1 hippocampal pyramidal neurons was reduced, as was the percentage of mature vs. immature spines in *Cdkl5* −/Y mice in comparison with control (+/Y) mice (Figure 6A–C). Treatment with **C11** restored the number of dendritic spines (Figure 6A,C) and the balance between immature and mature spines (Figure 6B,C) in *Cdkl5* −/Y mice.

Recruitment of PSD-95 to the postsynaptic compartment is one factor that contributes to the stabilization of synaptic contacts [40]. To confirm that the **C11**-induced restoration of spine maturation was accompanied by a restoration of synaptic connectivity, we evaluated the number of immunoreactive puncta for PSD-95 in the hippocampal of *Cdkl5* −/Y mice. As previously reported [11,41], in *Cdkl5* −/Y mice, the number of postsynaptic terminals was lower in comparison with that of *Cdkl5* +/Y mice (Figure 6D,E). Treatment with **C11** rescued this defect (Figure 6D,E).

### 2.6. Effect of Treatment with ***C11*** on Neuronal Survival in Cdkl5 KO Mice

Recent evidence has shown that, in addition to affecting synaptogenesis, *CDKL5* also impacts on neuronal survival [19,20,21]. *Cdkl5* KO mice are characterized by decreased survival of hippocampal neurons, which worsens with age [42].

In order to evaluate whether treatment with **C11** affects neuronal survival rate in *Cdkl5* KO mice, we evaluated cell density in the CA1 layer of the hippocampus. We found that *Cdkl5* −/Y mice showed a reduced number of Hoechst-positive nuclei and NeuN-positive pyramidal neurons in the CA1 layer in comparison with control (+/Y) mice (Figure 7A–C), confirming the reduced neuronal survival in the absence of *Cdkl5*. Importantly, treatment with **C11** rescued hippocampal neuronal survival (Figure 7A–C). Otherwise, in vivo treatment with NP12 [26] does not improve survival of pyramidal neurons in the hippocampus in *Cdkl5* −/Y mice (Appendix A), assessed as Hoechst-positive nuclear density in the CA1 layer.

### 2.7. Effect of Treatment with ***C11*** on Microglia Over-Activation in Cdkl5 KO Mice

It has been shown that **C11**, in addition to its dual GSK-3β/HDAC6 inhibitory action, decreases neuroinflammation by modulating the microglial phenotypic switch from the M1 (pro-inflammatory) to the M2 (anti-inflammatory) phenotype [29]. We recently found increased microglial activation in the brain of the *Cdkl5* KO mouse [43]; indeed, modulation of microglial activation seems to be a therapeutic approach to be considered for CDD. Similarly to previous findings, we found alterations in microglial cell morphology, with an enlarged body size that is typical of a state of activation [44], in the hippocampus of *Cdkl5* −/Y mice compared to their wild-type counterparts (Figure 7D,E). Interestingly, treatment with **C11** reversed the inflammatory status in *Cdkl5* −/Y mice, bringing microglial soma size to levels that were even lower than those of vehicle-treated control (+/Y) mice (Figure 7D,E).

## 3. Discussion

It is likely that, due to *CDKL5* failure, alterations in multiple complex molecular networks are involved in the mechanisms that underlie the CDD-related phenotype. The complexity of this disease might limit the efficacy of commonly used single molecular drug therapeutics. Therefore, the severity of the neurological phenotype of CDD patients could be better tackled using a multi-target approach that is able to simultaneously modulate multiple targets involved in the onset of the disease. Our study provides novel evidence that, in in vitro experimental models of CDD, the GSK-3β/HDAC6 dual inhibitor, **C11** [29], is more effective at recovering neuronal survival than treatment with a single inhibitor that is selective for GSK-3β. Importantly, in vivo treatment with **C11** restored synapse development, neuronal survival, and microglia over-activation, and improved motor and cognitive abilities of *Cdkl5* KO mice. Overall, our data suggest that a GSK-3β/HDAC6 dual inhibitor therapy may engender a more effective strategy with which to achieve therapeutic benefits in CDD patients.

To validate the effect of **C11** in CDD we selected, as an in vitro model, a recently generated human cellular model of *CDKL5* deficiency that exhibited alterations in the GSK-3β signal pathway [19] as well as reduced survival and differentiation. Here, we found that *CDKL5* deficiency in these cells causes decreased acetylation of a well-known substrate of HDAC6 deacetylase activity, αtubulin [45]. The decreased αtubulin acetylation, not resulting from deregulated HDAC6 expression (Appendix A), may be attributed to increased HDAC6 activity which, in turn, could be explained by the complex regulation of HDAC6 activity, including direct or indirect binding partners and kinase-mediated phosphorylation that leads to either increased or reduced HDAC6 activity [46]. Among the kinases that regulate HDAC6 activity, GSK-3β interacts and localizes with HDAC6, and enhances HDAC6 activity; thereby, it may indirectly influence tubulin acetylation [47]. The link between HDAC6 and GSK-3β may explain the partial recovery of tubulin acetylation in *CDKL5* deficient cells obtained with the single GSK-3β inhibitor, NP12.

Abnormal activation of GSK-3β has been associated with several neurological and psychiatric disorders that share developmental abnormalities and altered neurocircuitry maintenance, such as schizophrenia, bipolar disorder, autism, and Alzheimer’s disease (AD) [48,49,50]. The involvement of GSK-3β misregulation in a variety of brain abnormalities strongly supports its pivotal role in controlling basic mechanisms of neuronal function from brain bioenergetics to the establishment of neuronal circuits, modulation of neuronal polarity, migration, and proliferation [24]. In particular, the role of GSK-3β in phosphorylation of cytoskeletal proteins impacts neuronal plasticity, as cytoskeletal constituents are involved in the development and maintenance of neurites, and changes in the rate of stabilization/destabilization of microtubules could influence major cellular compartments of neurons, such as dendrites, spines, axons, and synapses. Confirming previous findings involving GSK-3β inhibitors [26,27], we found that pharmacological inhibition of the GSK-3β activity with **C11** improved or even restored neuronal maturation and connectivity in in vitro and in vivo models of CDD. In both human *CDKL5* deficient cells and hippocampal neurons from *Cdkl5* KO mice a complete restoration of the extension of the neuritic processes was obtained when cells were treated with either **C11** or NP12. Similarly, the effects of in vivo treatment with **C11** or NP12 were comparable in terms of dendritic spines and PSD-95 positive puncta in *CDKL5* KO mice (present findings and those of [41]).

Interestingly, we showed here that **C11** treatment was much more effective at recovering neuronal survival than treatment with NP12 in both in vitro and in vivo experimental models of CDD, suggesting that the **C11** pro-survival effect is conveyed to HDAC6 inhibition. In support of this, our findings are in line with a previous study showing that inhibition of HDAC6 activity in neurons promotes protection against oxidative stress-induced death [30]. There is a growing consensus that HDACs play a role in the regulation of neuronal survival [51,52,53]. Considerable research activity has focused on histone deacetylase (HDAC) inhibitors as neuroprotective agents for a number of neurodegenerative diseases and CNS injuries [52,54]. HDAC6, a member of Class IIb family of HDACs, regulates trafficking of neurotrophic factor, functions as an αtubulin deacetylase, and modulates mitochondrial transport in hippocampal neurons [55]. Importantly, HDAC6 is involved in several events of the neurodegenerative cascades [56]; specific HDAC6 inhibition exerts neuroprotection by increasing the acetylation levels of αtubulin with subsequent improvement of the axonal transport, which is usually impaired in neurodegenerative disorders [56]. HDAC6 is a specific deacetylase of the homodimeric molecular chaperone Hsp90, which sequesters the heat shock transcription factor 1 (HSF1) [57]. A further mechanism through which the inhibition of HDAC6 could exert its pro-survival action concerns the decreasing of HSP90 deacetylation. A number of reports suggest that HSP90 may be a viable target for neuroprotection, as HSP90 inhibitors have been found to promote HSF1 release and augment the heat-shock response, which protects against neurotoxic insults in a variety of models of neurodegenerative disease [58,59]. While we cannot rule out the possibility that other HDAC family members may also be targets for neuroprotection in *CDKL5* KO cells, our findings showing that treatment with **C11** selectively increases αtubulin acetylation, without altering histone H3 acetylation, strongly confirm the higher specificity of **C11** for HDAC6.

Evidence that microglia-mediated inflammation contributes to the neuropathology of CDD has recently been reported [43]. Chronic activation of microglia may cause reduced neuronal maturation and survival through the release of potentially cytotoxic molecules such as pro-inflammatory cytokines [60,61]. Therefore, suppression of microglia over-activation by **C11** might contribute to restoring neuronal survival and connectivity in the hippocampus of *Cdkl5* KO mice. The anti-inflammatory effect exerted by **C11** may depend on both of its inhibitory activities. GSK-3 inhibitors lead to a reduction in microglia activity [62,63] via increased expression of the inhibitory marker CD200R and diminished inflammatory mediators such as nitric oxide (NO), glutamate, pro-inflammatory cytokines (TNF-α and IL-6) [64]. The mechanism by which HDAC inhibition is anti-inflammatory is not understood, but inhibitors of HDACs reduce the inflammatory response of isolated microglia to stimulants such as lipopolysaccharide [65,66], and in vivo delivery of HDAC inhibitors reduces neuroinflammation in models of brain injury [67,68].

### Conclusions

The search for better results in clinical practices, because of the inefficacy of some treatments based on single drugs, has encouraged the adoption of polypharmacology as a new therapeutical strategy. This multi-target approach could be: association of drugs, combination of drugs, or a single drug with multiple ligands. However, the deleterious effects caused by drug–drug interactions and toxicity of combination therapy have reinforced the emergence of novel strategies in drug discovery. The most recent approach is now considering single drugs that concomitantly recognize more than one molecular target. Although GSK-3β and HDAC inhibitors have been previously tested in *Cdkl5* KO mice [11,26,27], this is the first combinatorial approach using a GSK-3β/HDAC6 dual inhibitor to study the beneficial effects of GSK-3β and HDAC6 inhibition in *CDKL5*-null neurons. The importance of the use of an inhibitor with large selectivity for HDAC6 such as **C11** is that it can overcome all of the neuronal toxicity associated with the use of pan-HDAC inhibitors, suggesting that its specific inhibition may eliminate a range of untoward effects seen with the clinical application of pan-HDAC inhibitors in cancer [69]. This is also consistent with the demonstration that mice lacking HDAC6 are viable and develop normally [45]. Our finding that treatment with **C11** improved motor and cognitive behavior in *Cdkl5* KO mice suggest that dual GSK-3β/HDAC6 inhibitors might be an effective treatment option for CDD. Since previous findings showed that a pharmacological intervention aimed at normalizing impaired GSK-3β activity elicits different age-dependent outcomes in *Cdkl5* KO mice [26], a multi-target compound, acting simultaneously on different pathways that are essential for neuronal maturation and survival, might have a better chance of being effective regardless of age. At present we do not know whether the therapeutic effect of the **C11** double inhibitor is more efficient at recovering brain abnormalities in CDD at different developmental ages than a therapy with a selective GSK-3β or HDAC6 inhibitor. Future studies may clarify this issue.

## 4. Materials and Methods

### 4.1. Compound ***11*** Synthesis

Compound **11** (**C11**) was synthetized as previously reported [29].

### 4.2. Cell Lines, Treatments and Measurements

Human neuroblastoma cell line SH-SY5Y, deriving from The European Collection of Authenticated Cell Cultures (Sigma-Aldrich, St. Louis, MO, USA) and the *CDKL5* knockout (KO) SH-SY5Y neuroblastoma cell line (SH-*CDKL5*-KO; [19]), were maintained in Dulbecco modified Eagle medium (DMEM, Thermo Fisher Scientific, Waltham, MA, USA) supplemented with 10% heat-inactivated FBS, 2 mM of glutamine, and antibiotics (penicillin, 100 U/mL; streptomycin, 100 μg/mL, Thermo Fisher Scientific, Waltham, MA, USA), in a humidified atmosphere of 5% of CO2 at 37 °C. Cell medium was replaced every 3 days and the cells were sub-cultured once they reached 90% confluence.

#### 4.2.1. **C11** and NP-12 Treatments

Cells were plated onto poly-D-lysine-coated slides in a 6-well plate at a density of 2.5 × 10^5^ cells per well in culture medium supplemented with 10% FBS. The day after, cells were exposed to **C11** (1 μM and 10 μM; stock solution 10 mM in 100% DMSO), NP-12 (Tideglusib; 1 μM; stock solution 10 mM in 100% DMSO; Sigma-Aldrich, St. Louis, MO, USA), or vehicle (0.1% DMSO) for 24 h.

#### 4.2.2. Hydrogen Peroxide Treatment

Cells were plated onto poly-D-lysine-coated slides in a 6-well plate at a density of 2.5 × 10^5^ cells per well in culture medium supplemented with 10% FBS. The day after, cells were exposed to H_2_O_2_ (200 μM; Sigma-Aldrich, St. Louis, MO, USA) and **C11** (10 μM) or NP12 (1 μM) for 24 h. The following day, cells were fixed in a 4% paraformaldehyde 4% glucose solution at 37 °C for 30 min.

#### 4.2.3. Retinoic Acid Induced Differentiation

For differentiation analyses, cells were plated onto poly-D-lysine-coated slides in a 6-well plate at a density of 1 × 10^5^ cells per well in culture medium supplemented with 10% FBS. Twenty-four hours after cell plating, retinoic acid (RA; Sigma-Aldrich, St. Louis, MO, USA) 10 mM in ethanol was added to the medium at 10 μM final concentration each day for 5 days. Cells were co-treated with **C11** (10 μM), NP12 (1 μM), or vehicle (0.1% DMSO) every 2 days.

#### 4.2.4. Analysis of Neurite Outgrowth

Phase contrast images of cells were taken with an Eclipse TE 2000-S microscope (Nikon Instruments Inc., Melville, NY, USAnikon) equipped with a DS-Qi2 digital SLR camera (Nikon Instruments Inc., Melville, NY, USA). Images were taken from random microscopic fields. Neurite outgrowth was measured using the image analysis system Image Pro Plus (Media Cybernetics, Silver Spring, MD, USA). Only cells with neurites longer than one cell body diameter were considered as neurite-bearing cells. In each experiment, a total of 900 cells was analyzed. All experiments were performed at least three times. The total length of neurites was divided by the total number of cells counted in the areas.

#### 4.2.5. Apoptotic and Mitotic Index

Nuclei were stained with Hoechst 33342 (Sigma-Aldrich, St. Louis, MO, USA). Apoptotic cell death was assessed by manually counting the number of pyknotic nuclei and apoptotic bodies and is expressed as a percentage of the total number of cells. The number of mitotic cells was assessed by manually counting the cells in prophase (chromosomes are condensed and visible), metaphase (chromosomes are lined up at the metaphase plate), and anaphase/thelophase (chromosomes are pulled toward and arrive at the opposite poles) and expressed as a percentage of the total number of cells.

### 4.3. Colony

The mice used in this work derive from the *Cdkl5* knockout (KO) strain in the C57BL/6N background developed in [13] and backcrossed in C57BL/6J for three generations. *Cdkl5* KO mice (*Cdkl5* −/Y) and age-matched controls (wild-type; *Cdkl5* +/Y), when possible littermates, were used for all experiments and genotyped as previously described [13]. The day of birth was designated as postnatal day (P) zero and animals with 24 h of age were considered as 1-day-old animals (P1). Mice were housed 3–5 per cage on a 12 h light/dark cycle in a temperature- (23 °C) and humidity-controlled environment with standard mouse chow and water ad libitum. The animals’ health and comfort were controlled by the veterinary service. All efforts were made to minimize animal suffering and to keep the number of animals used to a minimum.

### 4.4. Primary Hippocampal Cultures, Treatments and Measurements

Primary hippocampal neuronal cultures were prepared from 1-day-old (P1) wild-type (WT) and *Cdkl5* KO mice as described [70]. Briefly, hippocampi were dissected from mouse brains under a dissection microscope and treated with 2.5% trypsin (Sigma-Aldrich, St. Louis, MO, USA) for 15 min at 37 °C and 1% DNase (Sigma-Aldrich, St. Louis, MO, USA) for 2 min at room temperature before being triturated mechanically with a fire-polished glass pipette to obtain a single-cell suspension. Approximately 1.2 × 10^5^ cells were plated on coverslips coated with poly-L-lysine in 6-well plates and cultured in Neurobasal medium supplemented with B27 (Invitrogen, Thermo Fisher Scientific, Waltham, MA, USA) and glutamine (Invitrogen, Thermo Fisher Scientific, Waltham, MA, USA). Cells were maintained in vitro at 37 °C in a 5% CO2-humified incubator and fixed for immunostaining or western blot analysis on day 10 after plating (DIV10). Treatment was added to the conditioned medium and replenished every second day until DIV10. Cell cultures were fixed in a 4% paraformaldehyde 4% glucose solution at 37 °C for 30 min and processed for immunocytochemistry analysis.

#### 4.4.1. Morphological Analyses

To evaluate neuritic outgrowth and connectivity, fixed cells were stained with the following primary antibodies: a rabbit polyclonal anti-MAP2 (1:100, Merck Millipore, Burlington, MA, USA) and a mouse monoclonal anti-PSD-95 (1:100, Abcam, Cambridge, UK) antibody. Detection was performed using a FITC-conjugated anti-rabbit IgG (1:200, Jackson ImmunoResearch Laboratories, Inc., West Grove, PA, USA) and a Cy3-conjugated anti-mouse IgG (1:200, Jackson ImmunoResearch Laboratories, Inc., West Grove, PA, USA) antibody, respectively. Nuclei were counterstained with Hoechst-33342 (Sigma-Aldrich, St. Louis, MO, USA) and fluorescent images were acquired using a Nikon Eclipse Te600 microscope equipped with a Nikon Digital Camera DXM1200 ATI system (Nikon Instruments, Inc., Melville, NY, USA). The neuritic length of MAP2-positive neurons was measured and quantified by tracing along each neuronal projection using the image analysis system Image Pro Plus (Media Cybernetics, Silver Spring, MD, USA) as previously described [11]. The degree of synaptic innervation was evaluated by counting the number of PSD-95 positive puncta on proximal dendrites and expressed as the number of PSD-95 puncta per 20 μm of neuritic length. Immunofluorescence images were taken with a LEICA TCS SL confocal microscope (Leica Mycrosystems, Wetzlar, Germany). Fifty neurons for each condition were evaluated.

#### 4.4.2. Neuronal Survival

In order to assess survival of differentiated hippocampal neurons in culture, the number of MAP2-positive neurons was counted and expressed as a percentage of the total number of cells in culture evaluated through Hoechst staining.

### 4.5. In Vivo Treatment

**C11** was dissolved in a vehicle solution of 5.5% dimethyl sulfoxide (DMSO) in water.

#### 4.5.1. Acute **C11** Treatment

Two-month-old *Cdkl5* +/Y mice were treated with vehicle (5.5% of DMSO in water) and two-month-old *Cdkl5* −/Y mice were treated with vehicle or **C11** (10 mg/kg, 50 mg/kg or 100 mg/kg) administered intraperitoneally. Animals were sacrificed after 4 h.

#### 4.5.2. Chronic **C11** Treatment

Since in the mammalian brain, the dramatic rearrangements in structure and function that characterize childhood/adolescence result in a critical and sensitive period of brain development and suggest that early interventions would provide potential clinical benefits, we decided to chronically treat juvenile *Cdkl5* KO mice [71]. *Cdkl5* +/Y and *Cdkl5* −/Y male mice were randomly assigned to the experimental conditions and, starting from postnatal day 30 (P30), were treated with vehicle (5.5% of DMSO in water) or **C11** (50 mg/kg) administered via a daily intraperitoneal injection for 15 consecutive days. Body weight of the mice was monitored daily.

#### 4.5.3. Chronic NP12 Treatment

Brain sections that were processed for Hoechst staining, and used to evaluate hippocampal neuron survival, derived from animals used in [26]. Briefly, 3-week-old *Cdkl5* +/Y and *Cdkl5* −/Y male mice were treated with vehicle (corn oil; Sigma-Aldrich, St. Louis, MO, USA) or NP-12 (Tideglusib; 20 mg/kg body weight in corn oil; Sigma-Aldrich, St. Louis, MO, USA), administered via subcutaneous injection every other day for 20 days. The dose of NP12 was chosen based on [72].

### 4.6. Behavioral Assays

A total of 44 animals, from 9 litters, were used for behavioral studies. The sequence of the tests was arranged to minimize the effect of one test influencing subsequent evaluation of the next test, and mice were allowed to recover for 1–2 days between different tests. All behavioral studies and analyses were performed blind to genotype. Mice were allowed to habituate to the testing room for at least 1 h before the test, and testing was always performed at the same time of day. Three independent test cohorts were used. The first test cohort consisted of 15 animals (*Cdkl5* +/Y *n* = 1, *Cdkl5* −/Y *n* = 8, and *Cdkl5* −/Y + **C11**
*n* = 6) that were tested with the following assays: clasping, catalepsy, and Morris water maze. The second cohort consisted of 18 animals (*Cdkl5* +/Y *n* = 9, *Cdkl5* −/Y *n* = 3, and *Cdkl5* −/Y + **C11**
*n* = 6) that were tested with the following assays: clasping, catalepsy, and Morris Water Maze. The third cohort consisted of 11 animals (*Cdkl5* +/Y *n* = 5 and *Cdkl5* −/Y *n* = 6) that were tested with the following assays: clasping and catalepsy.

#### 4.6.1. Morris Water Maze

Hippocampal-dependent spatial learning and memory was assessed using the Morris Water Maze (MWM) as previously described [26]. Mouse behavior was automatically videotracked (EthoVision 3.1; Noldus Information Technology, Wageningen, the Netherlands). During training, each mouse was subjected to either 1 swimming session of 4 trials (day 1) or 2 sessions of 4 trials per day (days 2–5), with an intersession interval of 1 h (acquisition phase). Mice were allowed to search for the platform for up to 60 s. The latency to find the hidden platform was used as a measure of learning. Twenty-four h after the last acquisition trial, on day 6, the platform was removed and a probe test was run. Animals were allowed to search for the platform for up to 60 s. The latency of the first entrance into the former platform area and the average swim speeds were measured. A visual cue test was conducted after the probe test to assess sensorimotor ability, motivation, and visual ability. The percentage of floating was defined as the percentage of time swimming at a speed slower than 4 cm/s. Animals that failed the visual clue or floating test (1 *Cdkl5* +/Y, 1 *Cdkl5* −/Y and 2 *Cdkl5* −/Y + **C11**) were excluded from the analysis.

#### 4.6.2. Catalepsy Bar Test

The bar was set at a height of 6 cm. Mice were gently positioned, by placing both forelimbs on the bar and their hindlimbs on the floor. The time needed for the mice to remove both paws from the bar was measured using a stopwatch.

#### 4.6.3. Hind-Limb Clasping

Animals were suspended by their tail for 2 min and hind-limb clasping was assessed independently by two operators from video recordings. A clasping event is defined as the retraction of limbs into the body and toward the midline. The time spent hind-limb clasping was expressed as a percentage.

### 4.7. Western Blot Analysis

Total proteins from SH-SY5Y and SH-*CDKL5*-KO cell lines were lysates in ice-cold RIPA buffer (50 mM Tris–HCl, pH 7.4, 150 mM NaCl, 1% Triton-X100, 0.5% sodium deoxycholate, 0.1% SDS) supplemented with 1mM PMSF, and with 1% protease and phosphatase inhibitor cocktail (Sigma-Aldrich, St. Louis, MO, USA). Total protein from the hippocampus and cortex of 2-month-old *Cdkl5* −/Y and *Cdkl5* +/Y male mice were homogenized in ice-cold RIPA buffer supplemented with 1mM PMSF, and with 1% protease and phosphatase inhibitor cocktail (Sigma-Aldrich, St. Louis, MO, USA). Protein concentration for both cell and tissue extracts was determined using the Bradford method [73]. Equivalent amounts of protein (50 μg) were subjected to electrophoresis on a 4–12% Mini-PROTEAN^®^ TGX^TM^ Gel (Bio-Rad Laboratories, Inc., Hercules, CA, USA) and transferred to a Hybond-ECL nitrocellulose membrane (Amersham—GE Healthcare Life Sciences, Chicago, IL, USA). For acetylated tubulin detection, 10 μg of proteins were loaded on the gel. The primary and secondary antibodies used are listed in Appendix A. To quantify post-translational modifications, such as phosphorylation and acetylation, the nitrocellulose membranes were stripped in Restore Western Blot Stripping Buffer (Thermo Fisher Scientific, Waltham, MA, USA) for 20 min and reprobed with the corresponding total protein antibody. Repeated measurements of the same samples were performed by running from two to four different gels. The signal of one sample (internal control) was used to perform a relative analysis of the antigen expression of each sample on the same gel. We considered the control signal as 100 and assigned a value to the other sample as a percentage of the control. Data analysis was performed by averaging the signals obtained in two to four gels for each individual sample. The densitometric analysis of digitized Western blot images was performed using Chemidoc XRS Imaging Systems and Image Lab^TM^ Software (Bio-Rad Laboratories, Inc., Hercules, CA, USA). Images acquired with exposition times that generated protein signals out of a linear range were not considered for the quantification.

### 4.8. Histological and Immunohistochemistry Procedures

For histological and immunohistochemistry analysis, animals were euthanized with isoflurane (2% in pure oxygen) and sacrificed through cervical dislocation. Brains were quickly removed and cut along the midline. Right hemispheres were fixed via immersion in 4% paraformaldehyde in 100 mM phosphate buffer, pH 7.4, stored in fixative for 48 h, kept in 20% sucrose for an additional 24 h, and then frozen with cold ice. Subsequently, hemispheres were cut with a freezing microtome into 30 μm thick coronal sections that were serially collected in anti-freeze solution (30% glycerol; 30% ethylen-glycol; 10% PBS10X; 0.02% sodium azide; MilliQ to volume) and processed for immunohistochemistry procedures as described below. Left hemispheres were Golgi-stained as previously described [27]. All steps of sectioning, imaging, and data analysis were conducted blindly.

#### 4.8.1. NeuN and AIF-1 Immunohistochemistry and Measurements

One out of every 8 serial brain sections from the hippocampal formation was incubated with one of the following primary antibodies: mouse monoclonal anti-NeuN antibody (1:250; Merk Millipore, Burlington, MA, USA) or rabbit polyclonal anti-AIF-1 antibody (1:300; Thermo Fisher Scientific, Waltham, MA, USA). Sections were then incubated for 2 h at room temperature with a Cy3-conjugated anti-mouse secondary antibody (1:200, Jackson ImmunoResearch Laboratories, Inc., West Grove, PA, USA) or with a Cy3-conjugated anti-rabbit secondary antibody (1:200, Jackson ImmunoResearch Laboratories, Inc., West Grove, PA, USA). Nuclei were counterstained with Hoechst-33342 (Sigma-Aldrich, St. Louis, MO, USA). Fluorescent images were acquired using a Nikon Eclipse TE600 microscope equipped with a Nikon Digital Camera DXM1200 ATI System (Nikon Instruments Inc., Melville, NY, USA). The number of Hoechst-positive and NeuN-positive cells were manually counted using the Image Pro Plus software (Media Cybernetics, Silver Spring, MD, USA) and established as cells/mm^3^. Starting from 20X magnification images of AIF-1-stained hippocampal slices, AIF-1 positive microglial cell body size was manually drawn using the Image Pro Plus measurement function, and expressed in µm^2^.

#### 4.8.2. PSD-95 Immunohistochemistry and Measurements

One out of every 6 free-floating sections from the hippocampal formation was incubated with a rabbit polyclonal anti-PSD-95 antibody (1:1000, Abcam, Cambridge, UK) and with a CY3-conjugated anti-rabbit IgG secondary antibody (1:200, Jackson ImmunoResearch Laboratories, Inc. Laboratories, Inc., West Grove, PA, USA). For quantification of PSD-95 immunoreactive puncta, images from the stratum oriens of the hippocampus were acquired using a LEICA TCS SL confocal microscope (63X oil immersion objective, NA 1.32; zoom factor = 8 Leica Microsystems; Wetzlar, Germany). Three to four sections per animal were analyzed and puncta counts were performed on a single plan, 1024 × 1024 pixel images. Counting was carried out using Image Pro Plus software (Media Cybernetics, Silver Spring, MD, USA).

#### 4.8.3. Spine Density and Morphology

In Golgi-stained 100-μm-thick sections, spines of CA1 pyramidal neurons were counted using a 100× oil immersion objective lens. Dendritic spine density was measured by manually counting the number of dendritic spines on basal dendrites of CA1 pyramidal neurons. In each mouse, 15 dendritic segments (segment length: 10–30 μm) from each zone were analyzed and the linear spine density was calculated by dividing the total number of counted spines by the length of the sampled dendritic segment. The total number of spines was expressed per 10 μm. Based on their morphology, dendritic spines can be divided into five different classes which fall into two categories which also reflect their state of maturation (immature spines: filopodium-like, thin- and stubby-shaped; mature spines: mushroom- and cup-shaped). The total number of spines was expressed per μm and the number of spines belonging to each class was counted and expressed as a percentage.

### 4.9. Statistical Analysis

Data from single animals represented the unity of analysis. Results are presented as mean ± standard error (SE). Statistical analysis was performed using GraphPad Prism (version 7, San Diego, CA, USA). All datasets were analyzed using the ROUT method (Q = 1%) to identify significant outliers and the D’Agostino–Pearson omnibus test for normality. Datasets with normal distribution were analyzed for significance using Student’s t-test or a one-way analysis of variance (ANOVA) with genotype and treatment as factors. Post hoc multiple comparisons were carried out using the Fisher least significant difference (Fisher LSD) test. For the learning phase of the MWM test, statistical analysis was performed using a repeated three-way mixed ANOVA with genotype and treatment as grouping factors and day as a repeated measure. For categorical data, that is, percentages of spines, we used a chi-squared test. Animals identified as outliers by the use of Grubb’s test were excluded from the analyses [74]. A probability level of *p* < 0.05 was considered to be statistically significant.

## Figures and Tables

**Figure 1 ijms-22-05950-f001:**
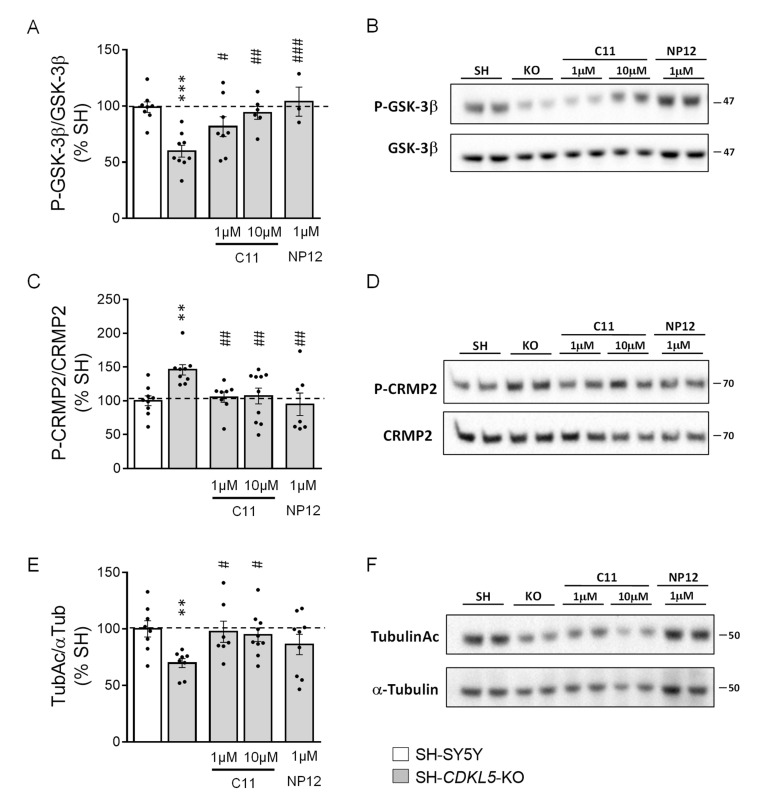
Effect of treatment with **C11** on GSK-3β and HDAC6 signaling in SH-*CDKL5*-KO cells. Western blot analysis of phospho-GSK-3β (Ser9; P-GSK-3β; **A**,**B**), P-collapsin response mediator protein 2-(Thr514P; P-CRMP2; **C**,**D**) and acetylated alpha tubulin (TubulinAc; **E**,**F**) levels in protein extracts from parental cells (SH-SY5Y; *n* = 8–9), SH-*CDKL5*-KO cells (SH-*CDKL5*-KO; *n* = 8–9) and SH-*CDKL5*-KO cells treated with **C11** (1 µM and 10 µM; *n* = 6–9) or NP12 (1 µM; *n* = 3–8) for 24 h. Immunoblots are examples from two biological replicates of each experimental condition. Histograms on the left show P-GSK-3β, P-CRMP2, and TubulinAc protein levels normalized to corresponding total protein levels. Data are expressed as a percentage of parental cells. Values are represented as means ± SE. ** *p* < 0.01; *** *p* < 0.001 as compared to the vehicle-treated SH-SY5Y condition; # *p* < 0.05; ## *p* < 0.01; ### *p* < 0.001 as compared to the vehicle-treated SH-*CDKL5*-KO condition (Fisher’s LSD test after one-way ANOVA).

**Figure 2 ijms-22-05950-f002:**
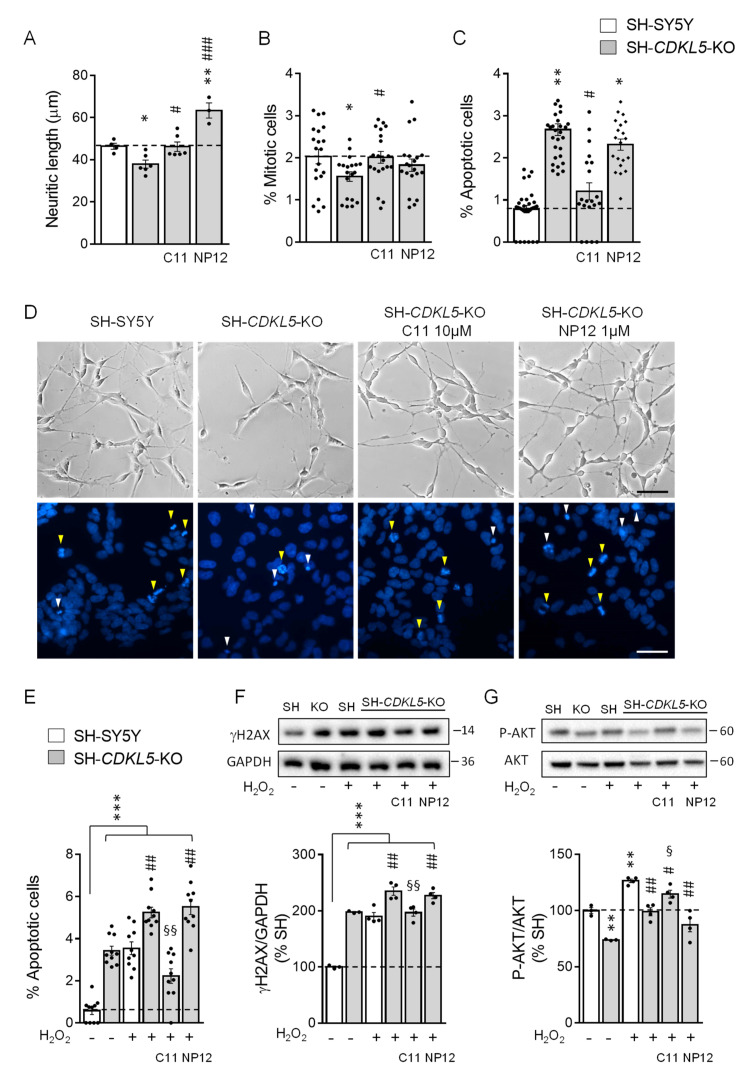
Effect of treatment with **C11** on cell differentiation, proliferation, and survival in SH-*CDKL5*-KO cells. (**A**) Quantification of neurite outgrowth of SH-SY5Y (*n* = 4) and SH-*CDKL5*-KO (*n* = 6) cells treated daily with retinoic acid (RA; 10 μM) for 5 days. SH-*CDKL5*-KO cells were treated with **C11** (10 µM, *n* = 6) or NP12 (1 µM, *n* = 6) every 2 days during retinoic acid differentiation. (**B**,**C**) Percentage of mitotic (**B**) and apoptotic (**C**) cells in proliferating SH-SY5Y (*n* = 20–30) and SH-*CDKL5* KO cells (*n* = 20–30); cells were treated with **C11** (10 µM, *n* = 20) or NP12 (1 µM, *n* = 20) for 24 h. (**D**) Representative phase-contrast images of neurite outgrowth (upper panel) of cells treated as in (**A**) (scale bar = 50 μm); fluorescence images of Hoechst-stained nuclei (lower panel; white triangles indicate pyknotic nuclei, yellow triangles indicate mitotic nuclei) of SH-SY5Y and SH-*CDKL5* KO cells treated as in (**B**,**C**) (scale bar = 30 μm). (**E**) Percentage of apoptotic cells in untreated (*n* = 10), and treated with H_2_O_2_ (200 μM, *n* = 10) for 24 h, parental SH-SY5Y and SH-*CDKL5*-KO cells, and in SH-*CDKL5*-KO cells co-treated with H_2_O_2_ (200 μM) and **C11** (10 µM, *n* = 10) or NP12 (1 µM, *n* = 10) for 24 h. (**F**,**G**) Western blot analysis of γH2AX (**F**) and phospho-AKT-Ser473 (P-AKT; (**G**)) levels in protein extracts of SH-SY5Y and SH-*CDKL5*-KO cells treated as in E (untreated cells *n* = 3; treated cells *n* = 4). Histograms show γH2AX protein levels normalized to GAPDH (**F**), and P-AKT levels normalized to total AKT protein levels (**G**). Examples of γH2AX, GADPH, P-AKT, and AKT immunoblots (upper panels). Values represent mean ± SE of 3 independent experiments. In (**A**–**C**) * *p* < 0.05; ** *p* < 0.01 as compared to the vehicle-treated SH-SY5Y condition; # *p* < 0.05; ### *p* < 0.001 as compared to the vehicle-treated SH-*CDKL5*-KO condition (Fisher’s LSD test after one-way ANOVA). In (**E**–**G**) ** *p* < 0.01; *** *p* < 0.001 as compared to the untreated SH-SY5Y cell condition; # *p* < 0.05; ## *p* < 0.001 as compared to the SH-SY5Y + H_2_O_2_ treated cell condition; § *p* < 0.05; §§ *p* < 0.001 as compared to the SH-*CDKL5*-KO + H_2_O_2_ treated cell condition (Fisher’s LSD test after one-way ANOVA).

**Figure 3 ijms-22-05950-f003:**
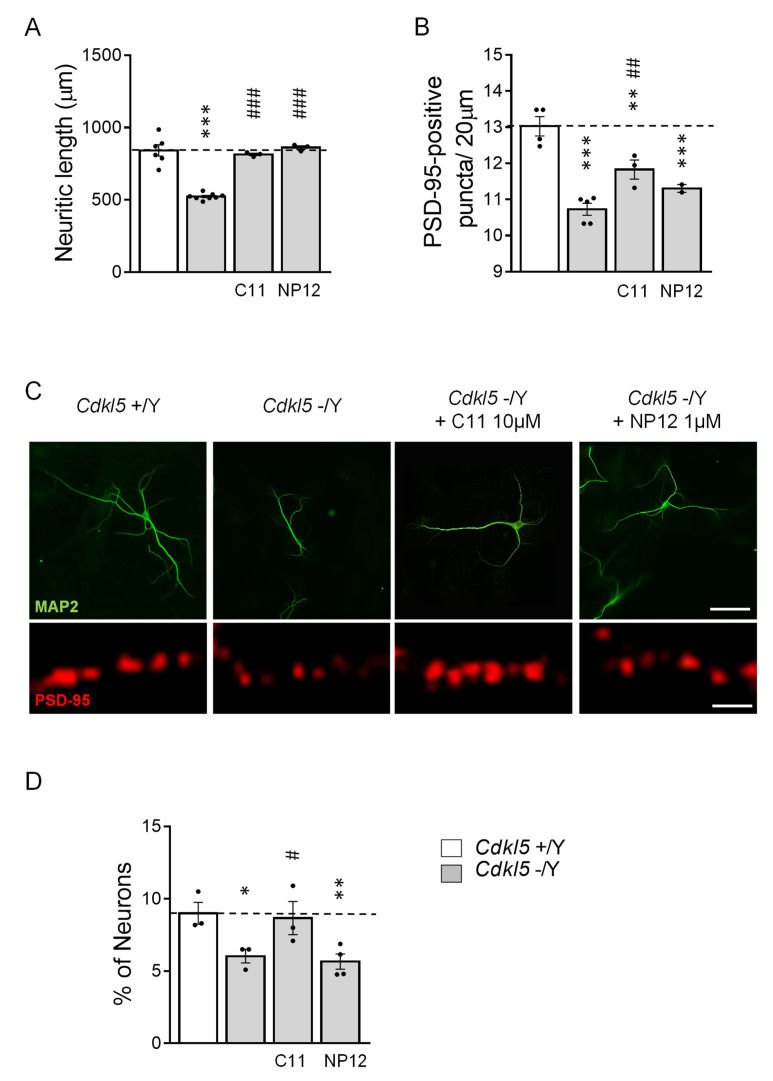
Effect of treatment with **C11** on maturation and survival of hippocampal neurons from *Cdkl5* KO mice. (**A**) Quantification of the total length of MAP2-positive cells from 10-day differentiated (DIV10) hippocampal neurons. On day 2 post-plating (DIV2) hippocampal cultures were treated with vehicle (0.1% DMSO in PBS; *Cdkl5* +/Y; *n* = 6 and *Cdkl5* −/Y; *n* = 8), **C11** (10 µM; *Cdkl5* −/Y + **C11**; *n* = 3), or NP12 (1 µM; *Cdkl5* −/Y + NP12, *n* = 3), which was then administered on alternate days throughout the entire differentiation period. (**B**) Quantification of the number of PSD-95 immunoreactive puncta per 20 μm in proximal dendrites of hippocampal neurons from vehicle-treated (*Cdkl5* +/Y; *n* = 4 and *Cdkl5* −/Y; *n* = 5), **C11**-treated (*Cdkl5* −/Y + **C11**; *n* = 3), and NP12-treated (*Cdkl5* −/Y + NP12; *n* = 2) hippocampal cultures. (**C**) Upper panels: representative fluorescence images of differentiated MAP2-positive hippocampal neurons from *Cdkl5* +/Y and *dkl5* −/Y mice treated as in (**A**) (scale bar = 40 µm). Lower panels show a magnification of a proximal dendrite immunopositive for PSD-95 (scale bar = 1 µm) of hippocampal neurons from *Cdkl5* +/Y and *Cdkl5* −/Y mice treated as in (**A**). (**D**) Quantitative analysis of the number of MAP2-positive cells in hippocampal cultures from wild-type (*Cdkl5* +/Y; *n* = 3), KO (*Cdkl5* −/Y; *n* = 3), KO treated with **C11** (*Cdkl5* −/Y + **C11**; *n* = 3), and KO treated with NP12 (*Cdkl5* −/Y + NP12; *n* = 4). Values are represented as mean ± SE. Values in D are represented as % of the *Cdkl5* +/Y conditions. * *p* < 0.05; ** *p* < 0.01; *** *p* < 0.001 as compared to the untreated cell *Cdkl5* +/Y condition; # *p* < 0.05; ## *p* < 0.01; ### *p* < 0.001 as compared to the untreated *Cdkl5* −/Y cell condition (Fisher’s LSD test after one-way ANOVA).

**Figure 4 ijms-22-05950-f004:**
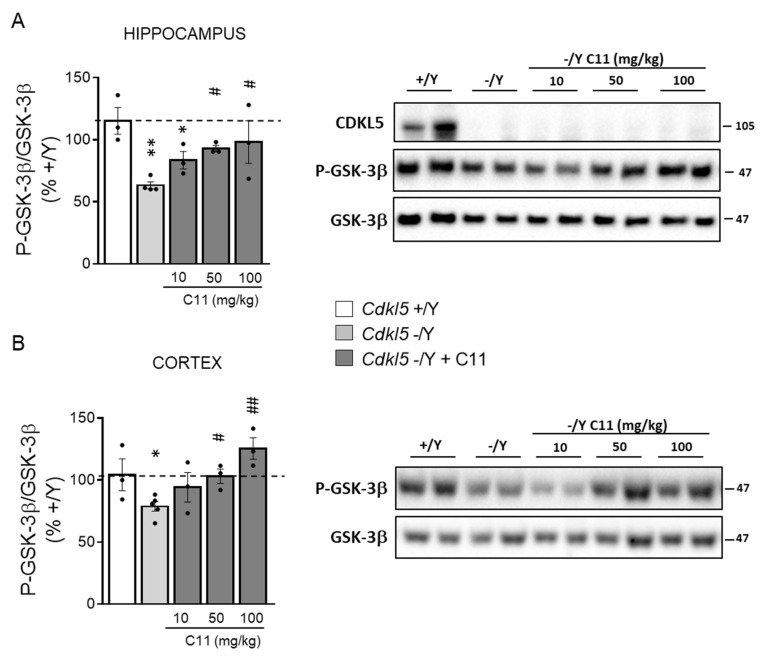
Effect of in vivo treatment with **C11** on GSK-3β activity in the brain of *CDKL5* −/Y mice. Western blot analysis of P-GSK-3β levels normalized to total GSK-3β levels in the hippocampus (**A**) and cortex (**B**) of wild-type (*Cdkl5* +/Y *n* = 3), KO (*Cdkl5* −/Y *n* = 4–5), and KO treated with a single intraperitoneal dose of **C11** (*Cdkl5* −/Y + **C11** 10 mg/kg *n* = 3, 50 mg/kg *n* = 3, 100 mg/kg *n* = 3) mice. Mice were sacrificed 4 h after the treatment. Histograms on the left show P-GSK3β protein levels normalized to corresponding total protein levels. Immunoblots on the right are examples from two animals of each experimental group of CDKL5, P-GSK-3β, and total GSK-3β levels. Values are represented as means ± SE and data are expressed as a percentage of untreated *Cdkl5* +/Y mice. * *p* < 0.05; ** *p* < 0.01 as compared to the untreated *Cdkl5* +/Y condition; # *p* < 0.05; ## *p* < 0.01 as compared to the untreated *Cdkl5* −/Y condition (Fisher’s LSD test after one-way ANOVA).

**Figure 5 ijms-22-05950-f005:**
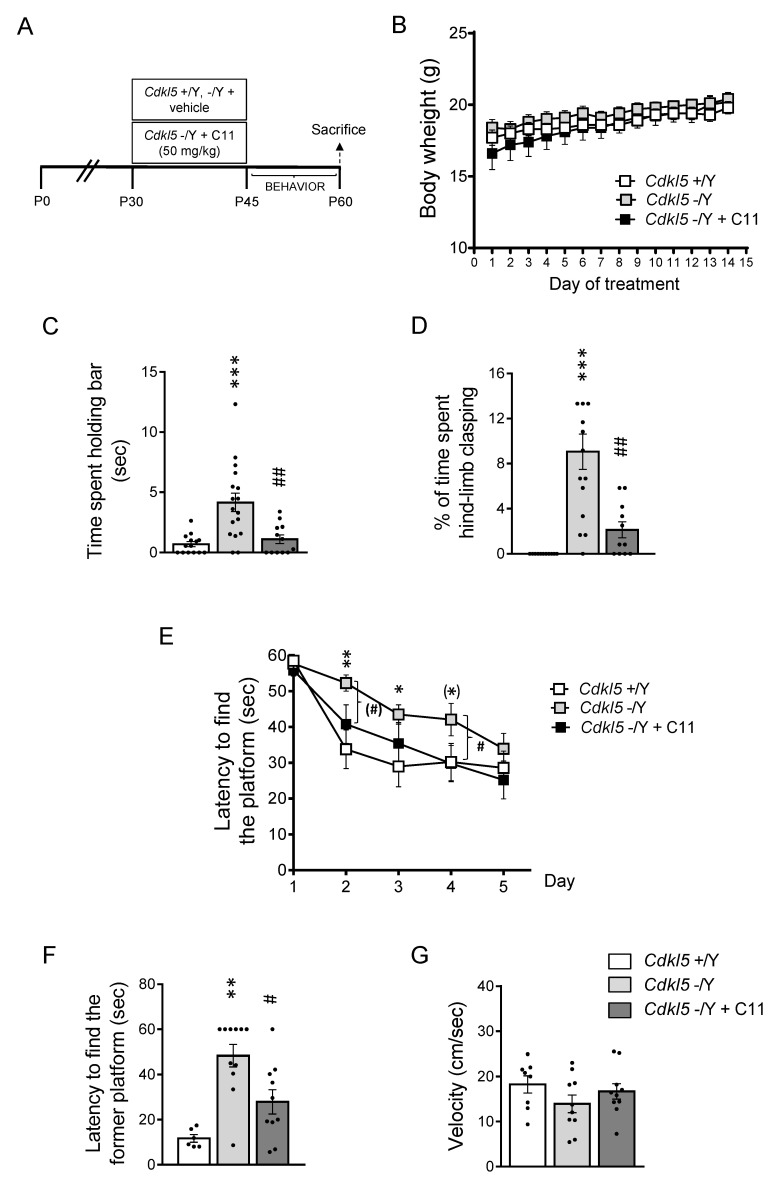
Effect of treatment with **C11** on motor coordination and hippocampus-dependent learning and memory in *Cdkl5* −/Y mice. (**A**) Experimental protocol. Starting from postnatal day 30 (P30), *Cdkl5* +/Y mice were treated with vehicle and *Cdkl5* −/Y male mice were treated with vehicle or **C11** (50 mg/kg), administered via intraperitoneal injection every day for 15 days. Animals from different experimental groups were behaviorally tested. Animals were sacrificed 15 days after the last injection (P60). (**B**) Body weight in grams of vehicle-treated *Cdkl5* +/Y (*n* = 15) and *Cdkl5* −/Y (*n* = 17) mice and **C11**-treated *Cdkl5* −/Y (*n* = 12) mice during the daily treatment period. (**C**) Amount of time taken to remove both paws from the bar in vehicle-treated (*Cdkl5* +/Y *n* = 14, *Cdkl5* −/Y *n* = 17) and **C11**-treated (*Cdkl5* −/Y + **C11**
*n* = 12) *Cdkl5* male mice. (**D**) Total amount of time spent hind-limb clasping during a 2 min interval in vehicle-treated (*Cdkl5* +/Y *n* = 14, *Cdkl5* −/Y *n* = 17) and **C11**-treated (*Cdkl5* −/Y + **C11**
*n* = 11) *Cdkl5* male mice. (**E**) Spatial learning (5-day learning period) assessed using the Morris Water Maze in vehicle-treated (*Cdkl5* +/Y *n* = 9, −/Y *n* = 10) and **C11**-treated (*Cdkl5* −/Y + **C11**
*n* = 10) *Cdkl5* male mice. (**F**) Spatial memory on day 6 (probe test) in mice as in (**D**). Memory was assessed by evaluating the latency to enter the former platform zone. (**G**) Mean velocity during the learning phase in mice as in (**D**). Values represent mean ± SE; (*) *p* = 0.058; * *p* < 0.05; ** *p* < 0.01; *** *p* < 0.001 as compared to vehicle-treated *Cdkl5* +/Y mice; (#) *p* = 0.059; # *p* < 0.05; ## *p* < 0.01 as compared to vehicle-treated *Cdkl5* −/Y mice. Dataset in (**B**) Fisher’s LSD after two-way ANOVA; dataset in (**E**) Fisher’s LSD after three-way mixed ANOVA; datasets in (**C**,**D**,**F**,**G**) Fisher’s LSD after one-way ANOVA.

**Figure 6 ijms-22-05950-f006:**
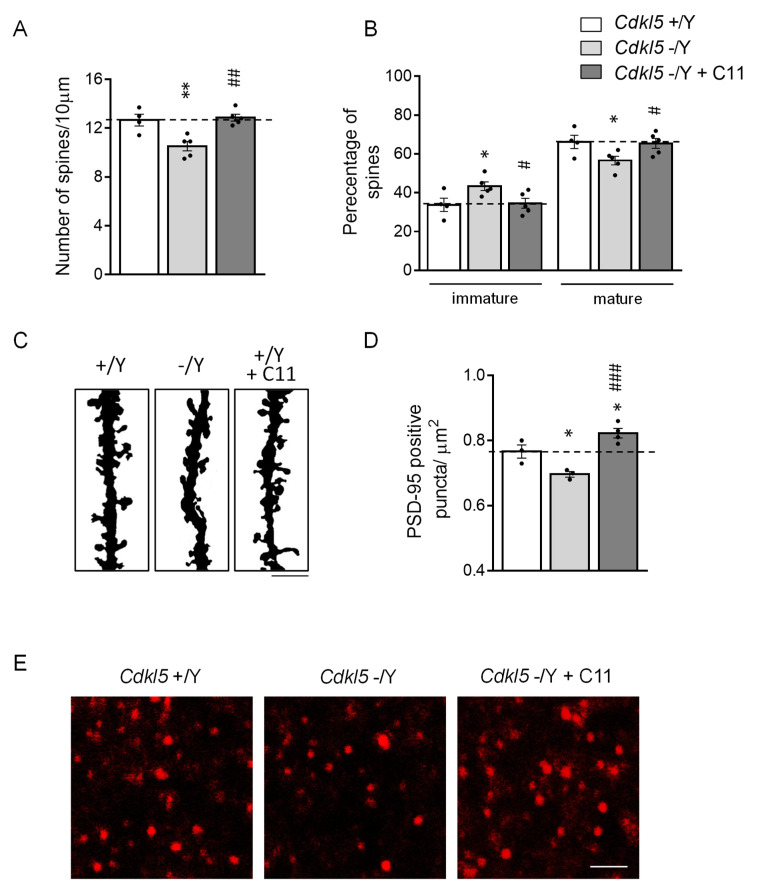
Effect of treatment with **C11** on dendritic spine density and maturation in the hippocampus of *Cdkl5* −/Y mice. (**A**) Dendritic spine density in CA1 pyramidal neurons of intraperitoneal vehicle-treated (*Cdkl5* +/Y *n* = 4, *Cdkl5* −/Y *n* = 5; P60) and **C11**-treated (*Cdkl5* −/Y *n* = 4; P60) *Cdkl5* male mice. Data are expressed as number of spines/10 μm. (**B**) Percentage of immature and mature spines in relation to the total number of protrusions in CA1 pyramidal neurons of mice as in A. (**C**) Examples of Golgi-stained dendritic branches of CA1 pyramidal neurons of one animal from each experimental group. Scale bar = 1 μm. (**D**) Number of fluorescent puncta per μm^2^ exhibiting PSD-95 immunoreactivity in the stratum oriens of the hippocampus of intraperitoneal vehicle-treated (*Cdkl5* +/Y *n* = 3; *Cdkl5* −/Y *n* = 3; P60) and **C11**-treated (*Cdkl5* −/Y *n* = 4; P60) *Cdkl5* male mice. (**E**) Representative confocal images of CA1-sections processed for PSD-95 immunohistochemistry of one animal from each experimental group. Scale bar = 2.5 μm. Values are represented as means ± SE. * *p* < 0.05; ** *p* < 0.01 as compared to the vehicle-treated *Cdkl5* +/Y condition; # *p* < 0.05; ## *p* < 0.01; ### *p* < 0.001 as compared to the vehicle-treated *Cdkl5* −/Y condition (datasets in (**A**,**D**), Fisher’s LSD after one-way ANOVA; dataset in (**B**), Dunn’s test after Kruskall–Wallis).

**Figure 7 ijms-22-05950-f007:**
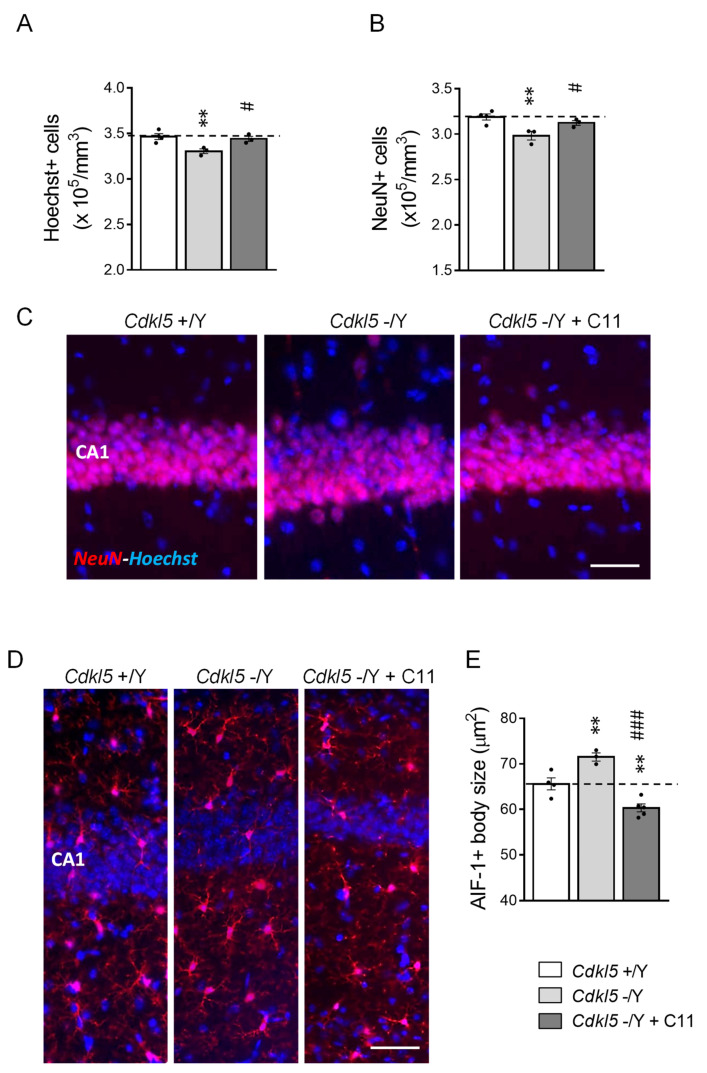
Effect of treatment with **C11** on neuronal survival and microglia activation in the hippocampus of *Cdkl5* −/Y mice. (**A**,**B**) Quantification of Hoechst-positive cells (**A**) and NeuN-positive cells (**B**) in CA1 layer of hippocampal sections from intraperitoneal vehicle-treated (*Cdkl5* +/Y; *n* = 4, *Cdkl5* −/Y; *n* = 5; P60) and **C11**-treated (*Cdkl5* −/Y + **C11**; *n* = 4; P60) mice. (**C**) Representative fluorescence images of immunopositive for NeuN (red) and counterstained with Hoechst (blue) sections in the hippocampal CA1 region of one animal from each group. Scale bar = 50 μm. (**D**) Representative fluorescence images of hippocampal sections processed for AIF-1 immunohistochemistry of one animal from each group. (**E**) Mean microglia cell body size in hippocampal sections from vehicle-treated (*Cdkl5* +/Y; *n* = 4, *Cdkl5* −/Y; *n* = 5; P60) and **C11**-treated (*Cdkl5* −/Y + **C11**; *n* = 4; P60) mice. Scale bar = 50 μm. Values are represented as means ± SE. ** *p* < 0.01, as compared to the vehicle-treated *Cdkl5* +/Y mice; # *p* < 0.05; ### *p* < 0.001 as compared to the vehicle-treated *Cdkl5* −/Y mice (Fisher’s LSD test after one-way ANOVA).

## Data Availability

The data that support the findings of this study are available upon request.

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
