# Peer review of "Treatment with a GSK-3β/HDAC Dual Inhibitor Restores Neuronal Survival and Maturation in an In Vitro and In Vivo Model of CDKL5 Deficiency Disorder"

_ijms, 2021, doi:10.3390/ijms22115950_

Round 1

Reviewer 1 Report

In their manuscript, Loi and colleagues tested the ability of the C11 compound, a GSK3beta/HDAC6 inhibitor, to rescue neuropathological and behavioural hallmarks in CDD. After reporting its efficacy in cellular models, they assessed its effectiveness in the Cdkl5 KO mice, as CDD animal model, indicating that targeting multiple molecular networks could be more efficacious.

The manuscript is well written and the rationale of the study is clearly described, as well as the results.

However I have some concerns regarding some methodological approaches and the methodological rigor in some experiments.

Initially the authors measure the inhibitor efficacy of C11, compared to NP12, in SH-SY5Y cells, by analysing n=2 samples for each experimental conditions. This means that data reported in the graphs in Figure 1A, C and E derive from the analysis of bands depicted in B, D and F, respectively. However, in some cases, data do not reflect the corresponding bands. See for example P-GSK3b/GSK3b C11 1uM.

If a stripping procedure has been used, it should be indicated in the Methods, reporting details.

In the Supplementary figures S1 it actually seems that the protein expression of HDAC6 is increased compared to control. Since n=2, I suppose that quantification derived from those bands reported in panel B. As minor modification, remove from the legend of Figure S1, the last sentence since there is no significant data.

Next, the authors reported C11 efficacy in ameliorating neurite length, proliferation and viability in differentiated SH-SY5Y. Neurite length in Figure 2A has to be presents as absolute values and not as percentage. The authors should describe in the Methods how they distinguished mitotic versus pyknotic nuclei, and in graph C data have to be reported as absolute values (as the authors did in E) and not as percentage. By analysing the C11 efficacy to protect from H2O2-induced stress, a group of untreated cells have to be included and the site of AKT phosphorylation has to be indicated.

When they analysed the effects on primary neurons, they reported an efficacy in improving neuronal outgrowth and synaptic defects. In Figure 3, data about neurite length have to be indicated as absolute values and considering that in C the images are representative for neuronal outgrowth, a minor magnification for each experimental conditions have to be added in order to appreciate the morphology of the entire neurons.

To support data obtained by counting neurons, the C11 efficacy on neuronal survival has to be assessed by a colorimetric assays, such as MTT. And in the Methods, more details about neuron culture preparations have to be reported. Did they added Ara-C? The presence of glial contamination has to be assessed by an immunofluorescence for GFAP.

Moving into an in vivo study, the authors administered in an acute experiment C11 to determine the efficient dose to be used in the next experiments, by analysing GSK3b phosphorylation as a read-out. Did the authors used the same filter to detect P-GSK3 and total GSK3b, after stripping? The shape of the bands corresponding to C11 100mg/Kg reveal that the filter is different. How do they normalize data?

A WB for CDKL5 in the same filters used for GSK3b should be added as a confirmation of the correct genotype.

The route of C11 administration should be indicated in the legend and not only in the text.

The first data reported is on body weight. However I suggest to replace Table 1, which reports only the weight at the beginning and the end of the treatment, with a graph reporting the weight of the animals along the treatment.

Then, in Figure 5, Loi and colleagues describe the behavioural outcome in sham and treated KO mice. A limit of the pharmacological study is the lack of the experimental group of WT treated mice, that would inform the reader on the side-effects of the drug and could allow for the proper statistical analysis of data by 2-way Anova. Further, it is not clear why the number of animals used throughout the different behavioural tests is different. Did the authors applied exclusion criteria? And the numbers of litters used to generate the experimental animals have to be indicated and if a randomisation has been applied.

The age of animals used for immunohistochemistry analysis has to be indicated in Figures 6 and 7.

In general, to better highlight the number of samples used and the internal variability, I suggest to modify all graphs, reporting data as scatterplots.

Author Response

-Initially the authors measure the inhibitor efficacy of C11, compared to NP12, in SH-SY5Y cells, by analysing n=2 samples for each experimental conditions. This means that data reported in the graphs in Figure 1A, C and E derive from the analysis of bands depicted in B, D and F, respectively. However, in some cases, data do not reflect the corresponding bands. See for example P-GSK3b/GSK3b C11 1uM.

ANSWER

We apologize to the Reviewer for not having clearly indicated how the Western blot experiments were performed and analyzed; this has now been specified in the Methods section (lines 683-691 of the revised version).

Images shown in Fig. 1B,D,F are examples of an immunoblot, but the Western blot analysis reported in Fig. 1 A,C,E derives from the quantification of several Western blots. Although Western blotting is a commonly-used technique in biological research, we found a great variability in our experiment. Our observation is in line with what was demonstrated by Aguilar and colleagues (Aguilar et al. 5(4):e9965 PlosOne, 2010). They performed an intra-assay and inter-assay variability analysis of Western blotting for GAPDH, loading 13 replicates of the same lysate in two SDS-PAGE mini-gels. They found a considerable intra-assay and inter-assay variability in signal intensity and concluded that, to obtain a reliable result, Western blotting has to be performed several times. For this reason, we replicated Western blotting analyses. We ran different gels (up to 4) for the two independent biological replicates. The signal of one sample (named internal control) was used to perform a relative analysis of the antigen expression in each sample on the same gel. We considered the control signal as 100 and assigned a value to the other sample as a percentage of the control. The densitometric analysis of digitized Western blot images was performed using Chemidoc XRS Imaging Systems and Image LabTM Software (Bio-Rad). The software automatically highlights any saturated pixels of the Western blot images in red, so we were able to avoid exposition times that generate protein signals out of a linear range.

-If a stripping procedure has been used, it should be indicated in the Methods, reporting details.

ANSWER

To quantify post-translational modifications, such as phosphorylation and acetylation, the nitrocellulose membranes were stripped in Restore Western Blot Stripping Buffer (Thermo Fisher) for 20 minutes and reprobed with the corresponding total protein antibody. We have now specified the stripping procedure in the Methods section (lines 679-683 of the revised version).

-In the Supplementary figures S1 it actually seems that the protein expression of HDAC6 is increased compared to control. Since n=2, I suppose that quantification derived from those bands reported in panel B. As minor modification, remove from the legend of Figure S1, the last sentence since there is no significant data.

ANSWER

Please see the above answer to the comment regarding Fig. 1. Western blot analysis reported in Figure S1 derives from the quantification of three different Western blots. We have replaced the immunoblot of Supplementary Fig. 1B with a more representative one (Figure S1B of the revised version). We have removed the sentence on significance from the legend of Figure S1.

-Next, the authors reported C11 efficacy in ameliorating neurite length, proliferation and viability in differentiated SH-SY5Y. Neurite length in Figure 2A has to be presents as absolute values and not as percentage. The authors should describe in the Methods how they distinguished mitotic versus pyknotic nuclei, and in graph C data have to be reported as absolute values (as the authors did in E) and not as percentage. By analysing the C11 efficacy to protect from H2O2-induced stress, a group of untreated cells have to be included and the site of AKT phosphorylation has to be indicated.

ANSWER

As requested by the Reviewer, in Fig. 2A and C of the revised version we have now reported the neurite length and the number of pyknotic nuclei as absolute values and not as percentages of control conditions.

The Reviewer is right, compaction of nuclear chromatin is a characteristic phenomena of both apoptotic execution and mitosis. However, while in mitosis the chromatin is condensed and reorganized into discrete chromosomes, during apoptosis the DNA attains a level of condensation that is even greater than that observed in mitosis (pyknotic nuclei are small condensed nuclei). Apoptosis results in morphological alterations of the cell nucleus, including pyknosis and cell shrinking. During chromatin condensation in apoptotic cells, extensive plasma membrane blebbing occurs, followed by karyorrhexis and separation of cell fragments into apoptotic bodies that consist of cytoplasm with tightly-packed organelles with nuclear fragments (Hendzel et al, JBC 273, 24470–24478, 1998; Zhao et al, Developmental Cell 36, 498–510, 2016). After staining with Hoechst 33342, condensed chromatin in chromosomes or apoptotic nuclei (pyknotic nuclei plus apoptotic bodies) are recognizable. We have changed the term “pyknotic nuclei” to “apoptotic cells” on the Y axis of the graphs of Fig. 2C,E. As requested by the reviewer, in the Methods section we have described how we distinguished mitotic from apoptotic nuclei (lines 547-555). Moreover, we have substituted the representative fluorescence images of Hoechst-stained nuclei of Fig. 2D with minor magnification images to include more cells and indicated apoptotic versus mitotic nuclei with colored arrows (Fig. 2D of the revised version).

As requested by the Reviewer, in the revised version we have included values of untreated SH-SY5Y and SH-CDKL5-KO cells in the graphs of Fig. 2E-G. An anti-phospho-AKT-Ser473 antibody was used in the Western blot analysis of Fig. 2G. This has now been specified in the legend of Figure 2.

-When they analysed the effects on primary neurons, they reported an efficacy in improving neuronal outgrowth and synaptic defects. In Figure 3, data about neurite length have to be indicated as absolute values and considering that in C the images are representative for neuronal outgrowth, a minor magnification for each experimental conditions have to be added in order to appreciate the morphology of the entire neurons.

ANSWER

As requested by the Reviewer, in Fig. 3A of the revised version we have reported the neurite length of hippocampal neurons as absolute values and not as percentages of the control condition. Moreover, we have substituted the representative fluorescence images of differentiated MAP2-positive hippocampal neurons of Fig. 3C with minor magnification images.

-To support data obtained by counting neurons, the C11 efficacy on neuronal survival has to be assessed by a colorimetric assays, such as MTT. And in the Methods, more details about neuron culture preparations have to be reported. Did they added Ara-C? The presence of glial contamination has to be assessed by an immunofluorescence for GFAP.

ANSWER

As requested by the Reviewer, in the revised version we have added more details on how primary hippocampal neuronal cultures were prepared following the protocol described in (Beaudoin et al. Nature Protocols 7, 2012). We have specified that hippocampi from 1-day-old wild-type (WT) and Cdkl5 KO mice were dissected from mouse brains under a dissection microscope and treated with trypsin for 15 min at 37°C and DNase for 2 min at room temperature before being triturated mechanically with a fire-polished glass pipette to obtain a single-cell suspension. Approximately 1.2 x 105 cells were plated on coverslips coated with poly-L-lysine in 6-well plates and cultured in Neurobasal medium supplemented with B27 and glutamine (lines 571-583 of the revised version).

The serum-free medium supplemented with B27 enhances neuronal survival, while it decreases glial cell proliferation to 2-3% (Beaudoin et al. Nature Protocols 7, 2012). Therefore, the addition of Ara-C is not necessary. The culture of postnatal versus embryonic neurons is advantageous because it reduces the necessity of killing animals including the mothers; however, the culturing of postnatal mouse neurons has proved to be more challenging than culturing embryonic mice, as it provides a lower yield of cultured neurons. Using this protocol in a differentiated neuronal culture (DIV10) we detected about 60-70% astrocytes, as identified by glial fibrillary acidic protein (GFAP) reactivity (please see supplementary information for Reviewer), and about 10-20% of neurons comprises inhibitory neurons.

Unfortunately, the colorimetric assay, MTT, is not reliable in our cell culture condition for two main reasons: i) due to the low percentage of neurons in culture the insoluble formazan crystals, which produce measurable color, are mainly due to the mitochondrial activity of astrocytes, masking the activity of the neuronal component and eventual differences between groups; ii) since in this assay it is not possible to standardize the number of cultured cells, the variability between groups is affected by the different number of plated cells, a difference that is frequent in the case of primary cultures. For these reasons, the assays that are not based on a direct count of the number of neurons (and expressed as a labeled index: number of neurons out of the total cell number), but on a indirect measurement of the activity of the total population, are highly variable from preparation to preparation and do not allow us to obtain comparable and reliable results.

-Moving into an in vivo study, the authors administered in an acute experiment C11 to determine the efficient dose to be used in the next experiments, by analysing GSK3b phosphorylation as a read-out. Did the authors used the same filter to detect P-GSK3 and total GSK3b, after stripping? The shape of the bands corresponding to C11 100mg/Kg reveal that the filter is different. How do they normalize data?

ANSWER

We apologize to the Reviewer for the mistake in the preparation of the panels of Figure 4. The Reviewer is right, the two upper immunoblots (P-GSK-3b and total GSK-3b in hippocampal extracts) do not derive from the same stripped and reprobed membrane, but from two different gels with the same samples. As stated in an above answer we ran different gels (in this case 3) for the independent biological replicates. We have now replaced the total GSK-3b immunoblot with the correct one (Fig. 4 of the revised version).

-A WB for CDKL5 in the same filters used for GSK3b should be added as a confirmation of the correct genotype.

ANSWER

We have now included an immunoblot showing CDKL5 levels in hippocampal extracts of the P-GSK-3b/GSK-3b immunoblots (Fig. 4 of the revised version).

-The route of C11 administration should be indicated in the legend and not only in the text.

ANSWER

As requested by the Reviewer, in each figure legend where the in vivo C11 treatment is reported we have now indicated that the route of administration was intraperitoneal.

-The first data reported is on body weight. However I suggest to replace Table 1, which reports only the weight at the beginning and the end of the treatment, with a graph reporting the weight of the animals along the treatment.

ANSWER

As suggested by the Reviewer, we have substituted Table 1 with a graph reporting the weight of the mice each day during treatment (Fig. 5B of the revised version).

-Then, in Figure 5, Loi and colleagues describe the behavioural outcome in sham and treated KO mice. A limit of the pharmacological study is the lack of the experimental group of WT treated mice, that would inform the reader on the side-effects of the drug and could allow for the proper statistical analysis of data by 2-way Anova. Further, it is not clear why the number of animals used throughout the different behavioural tests is different. Did the authors applied exclusion criteria? And the numbers of litters used to generate the experimental animals have to be indicated and if a randomisation has been applied.

ANSWER

The Reviewer is right, wild-type mice treated with C11 may provide information as to possible side-effects. To try to answer the Reviewer regarding the possible presence of toxic effects of C11 in the control conditions, in the revised manuscript we have added data regarding the in vitro effect of treatment with C11 in SH-SY5Y cells and primary hippocampal neurons. We found that in control SH-SY5Y cells and +/Y hippocampal neurons treatment with C11 did not have a negative effect on neuronal maturation and survival (Fig. S2 of the revised version).

Unfortunately, the synthesis of the compound C11, which is synthetized by the co-author Dr. Milelli, takes a long time to prepare, 4-6 months for a quantity that is only sufficient to chronically treat 6-8 mice. Therefore, for this study we decided to limit the treated experimental group to Cdkl5 -/Y mice in order to test the compound efficacy in ameliorating the pathological condition. We hope that this explanation satisfies the Reviewer, as it would be difficult to produce a sufficient amount of C11 for the in vivo experiment; the suggested experiment would take a long time to perform (at least 1 year), and would not fall within the usual time frame for a review.

We apologize with the Reviewer for not having specified the different cohorts of mice that were behaviorally tested; we also noticed some errors in the indication of the number of animals reported in the legend of Fig. 5, which have been now corrected. We have specified that Cdkl5 +/Y and Cdkl5 -/Y male mice were randomly assigned to the experimental conditions and, starting from postnatal day 30 (P30), were treated with vehicle (5.5% of DMSO in water) or C11 (50 mg/kg) administered via a daily intraperitoneal injection for 15 consecutive days.

A total of 44 animals, from 9 litters, were used for behavioral studies. The sequence of the tests was arranged to minimize the effect of one test influencing subsequent evaluation of the next test, and mice were allowed to recover for 1-2 days between different tests. All behavioral studies and analyses were performed blind to genotype. Mice were allowed to habituate to the testing room for at least 1 h before the test, and testing was always performed at the same time of day. Three independent test cohorts were used. The first test cohort consisted of 15 animals (Cdkl5 +/Y n = 1, Cdkl5 -/Y n = 8, and Cdkl5 -/Y + C11 n = 6) that were tested with the following assays: clasping, catalepsy, and Morris water maze. The second cohort consisted of 18 animals (Cdkl5 +/Y n = 9, Cdkl5 -/Y n = 3, and Cdkl5 -/Y + C11 n = 6) that were tested with the following assays: clasping, catalepsy, and Morris water maze. The third cohort consisted of 11 animals (Cdkl5 +/Y n = 5 and Cdkl5 -/Y n = 6) that were tested with the following assays: clasping and catalepsy (paragraph 4.6 Behavioral Assays of the revised version).

We have now indicated that animals that failed the visual clue or floating test in the MWM  (lines 655-657), as well as animals identified as outliers by the use of Grubb’s test, were excluded from the analyses (lines 754-755).

-The age of animals used for immunohistochemistry analysis has to be indicated in Figures 6 and 7.

ANSWER

For immunohistochemistry analysis we used the brain of mice sacrificed 15 days after the last injection at the end of the behavioral tests (P60; Fig. 4A). We have now specified the age of the mice in the legends of Figs. 6 and 7 of the revised version.

-In general, to better highlight the number of samples used and the internal variability, I suggest to modify all graphs, reporting data as scatterplots.

ANSWER

As suggested by the Reviewer, we have modified the graphs regarding behavioral results of Fig. 5, reporting data as scatterplots.

Reviewer 2 Report

The research paper entitled “Treatment with a GSK-3β/HDAC dual inhibitor restores neuronal survival and maturation in an in vitro 2
and in vivo model of CDKL5 deficiency disorder” is a very interesting and well written. The experimental studies were well and thoroughly planned and conducted. The introduction is comprehensive. The methods used are well chosen and described, and the results are clearly presented.

Author Response

We thank the Reviewer for the positive comment.

Reviewer 3 Report

In the paper titled “Treatment with a GSK-3β/HDAC dual inhibitor restores neuronal survival and maturation in an in vitro and in vivo model of CDKL5 deficiency disorder” by Loi et al. authors tested the ability of a first-in-class GSK-3β/HDAC dual inhibitor Compound 11 (C11) to rescue deficiency disorder (CDD). The paper is interesting.

I do have a few comments

What are the solution in which C11 and NP12 are diluted ? Pag 18 line 411

What is the solution in which 10 μM retinoic acid is diluted ? Pag 18 line 415

The authors need to explain why they choose the time and the dose of C11 for the Chronic C11 treatment. Similarly why did they choose time and dose for for NP-12 in the Chronic NP-12 treatment?

How many immunoblots were analysed for each experimental condition?

Line 194 the reference Bahi-Buisson et al. 2008 is not edited

Author Response

-What are the solution in which C11 and NP12 are diluted ? Pag 18 line 411

-What is the solution in which 10 μM retinoic acid is diluted ? Pag 18 line 415

ANSWER

As requested by the Reviewer, we have now indicated that from a stock solution of 10 mM in 100% DMSO,  C11 and NP12 were diluted in the culture medium  to a final concentration of  1 μM and 10 μM (C11) and 1 μM (NP-12). As a control, parallel cultures were treated with DMSO to reach a final concentration of 0.1% DMSO (lines 525-527). Retinoic acid 10 mM in ethanol was added to the culture medium at a 10 mM final concentration (lines 535-537).

-The authors need to explain why they choose the time and the dose of C11 for the Chronic C11 treatment. Similarly why did they choose time and dose for NP-12 in the Chronic NP-12 treatment?

ANSWER

In the mammalian brain, the dramatic rearrangements in structure and function that characterize childhood/adolescence result in a critical and sensitive period of brain development, and suggest that early interventions would provide potential clinical benefits. Our previous finding, that treatment with NP12 restores the major developmental defects that characterize the brain of Cdkl5 −/Y mice only when treatment is performed in an early time window of brain development P20-P40 (Fuchs et al. EJN, 2018), is in line with the potential benefits of early intervention. For this reason we decided to treated juvenile Cdkl5 KO mice, for two weeks (P30-P45), during brain development, before adulthood. According to the work of Finlay and Darlington (Science, 268, 1578-84, 1995) the date of postnatal day 50 is a reasonable standard for the definition of an adult mouse. The reason for choosing the age of the mice has been now indicated in the revised version (lines 613-617).

The dose of NP12 used in (Fuchs et al. Eur J Neurosci, 47, (9), 1054-1066,  2018; 20 mg/kg) was chosen based on (Zhou et al., Nature Cell Biology, 18, 2016). This has now been indicated in the revised version (line 627 of the revised version). Regarding the C11 dose, since there were no previous studies on the action of C11 in vivo, for chronic treatment in mice we selected the dose of C11 based on the pilot study shown in Fig. 4. We found that at the dose of 50 mg/kg, C11 increased cortical GSK-3β phosphorylation levels, bringing them to the values of those found in the Cdkl5 +/Y mouse. This higher dose of C11 needed to inhibit P-GSK3 to control levels in Cdk5 KO mice, in comparison with NP12 (Fuchs et al. Eur J Neurosci, 47, (9), 1054-1066,  2018), is consistent with the different in vitro efficacy of C11 and NP12 in inhibiting GSK-3β (C11: IC50 = 2.7 mM; NP12: IC50 = 60 nM).

-How many immunoblots were analysed for each experimental condition?

ANSWER

We found a considerable intra-assay and inter-assay variability in signal intensity and concluded that, to obtain a reliable result, Western blotting has to be performed several times. For this reason, we replicated Western blotting analyses. We ran at least three different gels for the independent biological replicates. This has now been specified in the Methods section (lines 679-691 of the revised version).

-Line 194 the reference Bahi-Buisson et al. 2008 is not edited

ANSWER

We thank the Reviewer for noting this; we have now included this reference.

Round 2

Reviewer 1 Report

Thanks the authors for responses to my comments.

I suggest to represent all data in all figures as scatterplot and to clearly indicate the number of samples used throught out the experiments in each figure legends.

Further, it is not clear to me why in the old versione of figure 3D the percentage of neurons in WT was 100% while in the new version is less than 10%, considering that no modification was made in the corresponding method of analysis and figure legend.

I see that astrocytes are present in these cellular cultures but other methods, except MTT, for neuronal survival exist that can better inform about neuronal health in culture. 

Author Response

-I suggest to represent all data in all figures as scatterplot and to clearly indicate the number of samples used throught out the experiments in each figure legends.

ANSWER

As suggested by the Reviewer, we have now modified all the graphs in the figures, reporting data as scatterplots, and have also indicated the number of samples in each figure’s legend.

-Further, it is not clear to me why in the old versione of figure 3D the percentage of neurons in WT was 100% while in the new version is less than 10%, considering that no modification was made in the corresponding method of analysis and figure legend.

ANSWER

We apologize to the Reviewer for not having clearly indicated how we expressed the data in the old version of figure 3D. In the former version, the neuronal number was expressed as a percentage of the control condition (Cdkl5 +/Y neurons), which were set to 100%. Differently, in the new version of figure 3D, we have expressed the number of neurons as a percentage of the total cells in culture.

-I see that astrocytes are present in these cellular cultures but other methods, except MTT, for neuronal survival exist that can better inform about neuronal health in culture. 

ANSWER

The Reviewer is right, there are other methods for labeling apoptotic neurons in culture, such as cleaved caspase-3 immunocytochemistry or Hoechst staining to visualize pyknotic nuclei. Indeed, in a previous study we found increased cleaved caspase-3-positive apoptotic death in differentiating hippocampal neurons (DIV 4) from Cdkl5 −/Y mice compared to control cultures (Fuchs et al. Brain Pathol. 2019 Sep;29(5):658-674), but not in differentiated (DIV 10) Cdkl5 -/Y hippocampal neurons. However, after 10 days of differentiation (DIV 10), Cdkl5 -/Y hippocampal cultures showed a lower number of neurons compared to controls, indicating that apoptotic cell death of Cdkl5 KO hippocampal neurons is a process that occurs mainly during differentiation. Since in our experimental paradigm treatment with C11 was prolonged for up to 10 days in culture (DIV10), we could only assess the number of viable neurons at the end of the treatment, as an indirect index of apoptotic cell death during differentiation.